# Diterpenes/Diterpenoids and Their Derivatives as Potential Bioactive Leads against Dengue Virus: A Computational and Network Pharmacology Study

**DOI:** 10.3390/molecules26226821

**Published:** 2021-11-11

**Authors:** Rasel Ahmed Khan, Rajib Hossain, Abolghasem Siyadatpanah, Khattab Al-Khafaji, Abul Bashar Ripon Khalipha, Dipta Dey, Umma Hafsa Asha, Partha Biswas, Abu Saim Mohammad Saikat, Hadi Ahmadi Chenari, Polrat Wilairatana, Muhammad Torequl Islam

**Affiliations:** 1Pharmacy Discipline, Life Science School, Khulna University, Khulna 9280, Bangladesh; raselahmed358@gmail.com; 2Department of Pharmacy, Life Science Faculty, Bangabandhu Sheikh Mujibur Rahman Science and Technology University, Gopalganj, Dhaka 8100, Bangladesh; khalipha1982@gmail.com (A.B.R.K.); hafsa4808@gmail.com (U.H.A.); 3Ferdows School of Paramedical and Health, Birjand University of Medical Sciences, Birjand 9717853577, Iran; asiyadatpanah@yahoo.com (A.S.); ahmadi.h@bums.ac.ir (H.A.C.); 4Department of Medical Laboratory Technology, Al-Nisour University College, Baghdad 10001, Iraq; k.a.alkhafaji@gmail.com; 5Department of Biochemistry and Molecular Biology, Life Science Faculty, Bangabandhu Sheikh Mujibur Rahman Science and Technology University, Gopalganj, Dhaka 8100, Bangladesh; diptadey727@gmail.com (D.D.); asmsaikat.bmb@gmail.com (A.S.M.S.); 6Department of Genetic Engineering and Biotechnology, Faculty of Biological Science and Technology, Jashore University of Science and Technology (JUST), Jashore 7408, Bangladesh; partha160626@just.edu.bd; 7Department of Clinical Tropical Medicine, Faculty of Tropical Medicine, Mahidol University, Bangkok 10400, Thailand

**Keywords:** *Aedes aegypti*, dengue virus, diterpene, molecular docking, NS5, NS1

## Abstract

Dengue fever is a dangerous infectious endemic disease that affects over 100 nations worldwide, from Africa to the Western Pacific, and is caused by the dengue virus, which is transmitted to humans by an insect bite of *Aedes aegypti.* Millions of citizens have died as a result of dengue fever and dengue hemorrhagic fever across the globe. Envelope (E), serine protease (NS3), RNA-directed RNA polymerase (NS5), and non-structural protein 1 (NS1) are mostly required for cell proliferation and survival. Some of the diterpenoids and their derivatives produced by nature possess anti-dengue viral properties. The goal of the computational study was to scrutinize the effectiveness of diterpenoids and their derivatives against dengue viral proteins through in silico study. Methods: molecular docking was performed to analyze the binding affinity of compounds against four viral proteins: the envelope (E) protein, the NS1 protein, the NS3 protein, and the NS5 protein. Results: among the selected drug candidates, triptolide, stevioside, alepterolic acid, sphaeropsidin A, methyl dodovisate A, andrographolide, caesalacetal, and pyrimethamine have demonstrated moderate to good binding affinities (−8.0 to −9.4 kcal/mol) toward the selected proteins: E protein, NS3, NS5, and NS1 whereas pyrimethamine exerts −7.5, −6.3, −7.8, and −6.6 kcal/mol with viral proteins, respectively. Interestingly, the binding affinities of these lead compounds were better than those of an FDA-approved anti-viral medication (pyrimethamine), which is underused in dengue fever. **Conclusion:** we can conclude that diterpenoids can be considered as a possible anti-dengue medication option. However, in vivo investigation is recommended to back up the conclusions of this study.

## 1. Introduction

Dengue (pronounced Den’gee) is a viral disease caused by one of the dengue virus strains, namely DEN1, DEN2, DEN3, and DEN 4 [1,2]. Viral transmission to humans occurs by the infected mosquito bite of an *Aedes aegypti* type. Dengue virus (DENV) is an RNA virus, otherwise known as arboviruses, and belongs to the Flaviviridae family [3]. The DENV genome has 11,000 nucleotide bones. They have three different protein molecules, C, prM, and E, that form virus particle. They also contain seven other types of protein molecules (NSI, NS2a, NS2b, NS3, NS4a, NS4b, and NS5) found in infected host cells and are instrumental for viral replication [1].

DENV is an enveloped, single-stranded, positive-sense virus with a 10.7 kb RNA genome [4,5], which is translated as a single polyprotein and then cleaved into three structural proteins, e.g., capsid (C), remembrance/membrane (prM/M), and envelope (E) and seven non-structural (NS) proteins, by a virus- and host-encoded proteases. The three structural proteins are important for capsid formation (C) and assembly into viral particles (prM and E). The non-structural proteins contain a serine protease and ATP-dependent helicase (NS3), which is required for virus polyprotein processing, a methyltransferase, and RNA-dependent RNA polymerase (NS5), and a cofactor for the NS3 protease (NS2B). NS4B has been implicated in blocking the interferon (IFN) response. NS1, NS2A, and NS4A have unknown or incompletely understood functional activities of dendritic cell-specific intercellular adhesion molecule-3-grabbing non-integrin (DC-SIGN) on dendritic cells [6], followed by viral uptake by receptor-mediated endocytosis. Endosomal acidification causes fusion of the viral and endosomal membranes and nucleocapsid. These NS proteins may be necessary for replication. In primary infection, the virus (DENV) enters target cells by the adherence of E protein to cell surface receptors, such as release into the cytoplasm [7,8]. Virus genome replication occurs in discrete domains within the endoplasmic reticulum (ER), assembly occurs in the ER, and virions come out via Golgi-derived secretory vesicles [9].

After being bitten by a mosquito of *A. aegypti* or *A*. *albopictus* type [10], DENV can cause a range of mild-to-severe illnesses. Each year it is believed to infect 50 to 100 million people worldwide (https://www.who.int/denguecontrol/epidemiology/en/, accessed on 20 June 2021), having a mortality rate of 1–5% without treatment and less than 1% with treatment. Severe illness (dengue hemorrhagic fever, D. S. S) has a mortality rate of 26%. The dengue rate has climbed to a 30-fold increase between 1960 and 2010. The reason is believed to be factors such as rapid urbanization, population growth, international travel from endemic areas, and global warming. The geographical area around the equator, mainly Asia and the Pacific, is mostly affected [1]. Initial global estimates of total dengue infections were based on an assumption of a constant annual infection rate among a crude approximation of the population at risk (10% of one billion [10] or 4% of two billion [11], resulting in several 80–100 million infections per year worldwide in 1988 [10,11]. Later, with more information regarding the ratio of dengue hemorrhagic fever to dengue fever cases, and the ratio of deaths to dengue hemorrhagic fever cases, the global infection was revised to 50–100 million infections per year [12,13,14,15]. 

In research by da Silva and his co-workers, it is indicated that labdane diterpenes isolated from the oil-resin fractions of *Copaifera reticulata* exhibit considerable larvicidal activity (median lethal concentration (LC_50_ = 0.8 ppm) towards the mosquito [16]. Labdane diterpenoid of the same species (*C. reticulata*) caused the death of *A. aegypti* larvae by cell destruction in the midgut [17]. The diterpene content of the essential oil of *Lantana montevidensis* also showed a larvicidal effect against *A. aegypti* [18]. Methyl dodovisate A and B, isolated from the aerial parts of *Dodonaea viscose*, is reported to exhibit a larvicidal effect (LC_50_ > 30 μg/mL) on *A. albopictus* larvae [19]. On the other hand, caesalacetal and caesaljapin isolated from *Sulcobrachus sauteri* also displayed anti-*A. albopictus* activity with LC_50_ values of 3 and 9 μg/mL, respectively [20]. The lipophilic nature of diterpenes and their derivatives enables them to cross the biological membranes which include the blood-brain barrier (BBB). Additionally, they can induce oxidative stress, increase levels of nitric oxide (NO) in the infected host, reduce the anti-viral resistance to reactive oxygen species (ROS), increase lipid peroxidation, and induce chronic inflammatory responses and cell membrane damage [21].

As a result, there are two major strategies for developing new DENV agents. To begin, the compound must (i) precisely inhibit the host behavior involved in viral replication while not affecting the cell’s normal function, and (ii) be able to adequately inhibit the host factor in vivo throughout physiological conditions [22]. Some of the natural diterpenes/diterpenoids and their derivatives were shown to exert a prominent effect on DENV vectors and exhibit cytotoxic effects on DENV as well. Moreover, these diterpenes/diterpenoids exert their anti-viral viral effects through different mechanisms of action, including the anti-DENV effect and larvicidal activity [23]. In this regard, this research aimed to look into the in silico ability of diterpenoids and their derivatives against the proteins that make up viral proteins.

## 2. Results and Discussion

### 2.1. Attribution of Proteins’ Active Sites and Validation

The binding sites of receptor proteins of dengue virus envelope (E) protein, NS3, NS5, and NS1 were predicted through the CASTp server using default parameters of the webserver [24]. In envelope (E) protein has 74 binding pockets that were characterized to attain residues probe radius 1.4 Å. Moreover, NS3, NS5, NS1. The amino acid residues involved in the conformation of binding pockets are depicted in Figure 1.

### 2.2. Computational Virtual Screening of Diterpenoids and Their Derivatives

#### ADMET Analysis

For the analysis and optimization of pharmacokinetic properties, the pkCSM and Swiss ADME approach confer a platform. The ADMET study of diterpenoids and their derivatives are shown in Table 1 and Figure 2. 

### 2.3. Molecular Docking

#### 2.3.1. Docking Approach of Natural Bioactive against DENV Receptor Proteins

##### Interaction with Viral Envelope (E) Protein

The best docking energy for the viral envelope (E) protein is the natural ligands triptolide, stevioside, alepterolic acid, and sphaeropsidin A with the binding energies of −8.1, −8.4, −8.3, and −8.7 kcal/mol, respectively. The binding mode study was carried out on the next four active compounds, and the results are shown in Table 2. Additionally, the existence of hydrogen bonds between the phytochemicals and the viral E protein stabilizes the ligand within its binding locations. The docking complexes were visually inspected in-depth for the interactions and binding mechanisms of each ligand with the functional residues of the DENV E protein (Figure 3).

Triptolide, a component of the medicinal plant *Tripterygium wilfordii* Hook, displays energy and is known to be useful against a variety of diseases, including lupus, cancer, rheumatoid arthritis, and nephrotic syndrome [25,26]. Triptolide has been demonstrated to suppress dengue reproduction [27], HIV1 replication [28], and herpes virus viral titer in recent research [29]. At 0.5–4 nM, it (triptolide) showed anti-DENV activity in a DENV model [27]. Whereas stevioside is a natural sweetener [30], *Stevia rebaudiana* displayed −9.3 kcal/mol against NS1 proteins and exhibited an anti-rota viral effect in combination with *Sophora flavescens* plant extract [31]. Along with the anti-viral effect of stevioside demonstrated anti-inflammatory effect [32], anti-hyperglycemic effect [33], and so on.

Furthermore, sphaeropsidin A, a fungal metabolite (phytotoxin), was found from *Diplopia cupressi*, which has a larvicidal effect (LD_50_: 36.8 ppm) on *A. aegypti* [34]. Moreover, sphaeropsidin A has the potential ability to include anti-biofilm activity, anti-microbial activity [35], and anti-cancer activity [36]. In our molecular docking study, sphaeropsidin A displayed good binding energy with DENV NS1 receptor protein through two hydrogen bonds and some other conventional hydrogen bonds, pi-pi, pi-alkyl bonds (Table 2).

Alepterolic acid is an ent-labdane diterpene found as a major metabolite from *Aleuritopteris argentea* (S. G. Gmél.) Fée is a medicinal fern. Alepterolic acid exhibited dengue larvicidal properties with an LC_50_ of 87.3 ppm. Additionally, it has shown potential selectivity towards *Trypanosoma brucei* with a median inhibitory concentration (IC_50_) of 3.42 μM [37]. Incorporation of the amino moiety into alepterolic acid can inhibit the proliferation of the cervical cancer cell line HeLa and induce apoptosis through the mitochondrial pathway [38].

##### Interaction with Viral NS3

Natural ligands stevioside, sphaeropsidin A, methyl dodovisate A, and caesalacetal showed the best binding energies for the viral NS3 protein with −8.0, −8.3, −9.2, and −8.0 kcal/mol, respectively. The binding mode study was carried out on the next four active compounds, and the results are shown in Table 3. The existence of hydrogen bonds between the phytochemical and the viral E protein additionally stabilizes the ligand within its binding locations. The docking complexes were visually inspected in-depth for the interactions and binding mechanisms of each ligand with the functional residues of the DENV E protein (Figure 4).

Stevioside is a natural sweetener [30]; *Stevia rebaudiana,* displayed −9.3 kcal/mol against NS1 proteins and showed inhibitory activity against NS2B-NS3pro of DENV4, with IC50 values of 14.1 ± 0.2, 24.0 ± 0.4, and 15.3 ± 0.4 µg/mL, respectively, where it is present in a mixture or similar compounds such as rebaudioside A (Reb-A), or steviol glycosides (SG), etc. [31,32]. It also has been associated with anti-hyperglycemic properties [33], and so on. Sphaeropsidin A was also found to have a larvicidal impact on Aedes aegypti (LD50: 36.8 ppm) [34]. Furthermore, anti-biofilm activity, antibacterial activity [35], and anti-cancer activity are all possible with sphaeropsidin A. [36]. Sphaeropsidin A showed good binding energy with dengue viral NS1 receptor protein in molecular docking research, thanks to two hydrogen bonds and additional traditional hydrogen bonds, pi–pi, and pi–alkyl bonds (Table 3).

On the other hand, methyl dodovisate A is isolated from the aerial parts of *D. viscosa*. It showed a larvicidal effect with an LC_50_ > 30 μg/mL on *A. albopictus* [38]. Furthermore, caesalacetal, isolated from *S. sauteri* also displayed anti-*A. albopictus* activity with LC_50_ values of 3 μg/mL [20].

##### Interaction with Viral NS5

With DENV protein NS5, phytochemicals, triptolide, stevioside, andrographolide, and caesalacetal demonstrated good to moderate binding energies of −8.8, −9.4, −8.4, and −8.4 kcal/mol, respectively (Table 4). The existence of hydrogen bonds between the phytochemical and the viral NS5 protein additionally stabilizes the ligand within its binding locations. The docking complexes were visually inspected in-depth for the interactions and binding mechanisms of each ligand with the functional residues of the DENV protein (Figure 5).

Andrographolide is a lactone diterpene, isolated from *Andrographis paniculata,* and possesses many biological effects, including antioxidant [39], anti-inflammatory [40], neuroprotective [41], hepatoprotective [42], anti-viral [43,44,45], anti-thrombotic [46], anticancer [47], and others. This diterpene lactone at 100 and 200 μM concentration showed an anti-DENV effect via GRP78 interaction pathway and at 5, 10, 15, and 25 ppm, a concentration larvicidal effect by inducing cytopathic effects in the midgut epithelium (LC_50_: 12 ppm) [23].

##### Interaction with Viral NS1

The natural ligands triptolide, stevioside, sphaeropsidin A, and caesalacetal have the best docking energy for the viral NS1 protein, with binding energies of −8.3, −9.3, −8.5, and −8.5 kcal/mol, respectively. The binding mode study was carried out on the next four active compounds, and the results are shown in Table 5. Furthermore, the existence of hydrogen bonds between the NS1 receptor protein and the phytochemical stabilizes the ligand in its binding locations. By visually inspecting the docking complexes, the interactions and binding mechanisms of each ligand with the functional residues of the DENV NS1 protein were investigated in-depth (Figure 6).

Triptolide, a component of the medicinal plant *Tripterygium wilfordii* Hook, displays energy and is known to be useful against a variety of diseases, including lupus, cancer, rheumatoid arthritis, and nephrotic syndrome [26,48]. Triptolide has been demonstrated to suppress DENV reproduction [27], HIV1 replication [28], and herpes virus viral titer in recent research (Long et al., 2016) [49]. At 0.5-4 nM, it showed anti-DENV activity in a DENV model [27]. On the other hand, stevioside displayed −9.3 kcal/mol against NS1 proteins and exhibits an anti-rota viral effect in combination with *S. flavescens* plant extract [31]. Along with its anti-viral effect, it also demonstrated an anti-inflammatory effect [32], anti-hyperglycemic effect [33], and so on.

Besides its larvicidal effect [34], sphaeropsidin A possess the potential ability to include anti-biofilm, anti-microbial [35], and anti-cancer activity [36]. In our molecular docking study, this gamma-lactone fungal metabolite displayed good binding energy with DENV NS1 receptor protein through two hydrogen bonds and some other conventional hydrogen bonds, pi-pi, pi-alkyl bonds (Table 6).

Caesalacetal, a cassane-type furanoditerpenoids, is mostly found in *S. sauteri* [20]. It is also isolated from the roots of *C. decapetala* var [50]. It exhibited larvicidal activities with an LC_50_: 3 μg/mL in the DENV vector [20]. It further demonstrated anti-viral activity against the protein NS1 (Table 5). The 2D and 3D structures of non-bond interactions of triptolide, stevioside, sphaeropsidin A, and caesalacetal with the target protein NS1 are shown in Figure 6.

#### 2.3.2. Docking Approach of Chemical Analog (Pyrimethamine) against DENV Proteins

The chemical compound (pyrimethamine), a DENV NS2B/3 protease inhibitor that has been shown to impede DENV translation and polyprotein processing [51], specifically at one intramolecular cleavage site within NS3 [52]. In molecular docking study, pyrimethamine has demonstrated good binding energies with four DENV receptor proteins E protein, NS3, NS5, and NS1 (Table 6) to be −7.5, −6.3, −7.8, and −6.6 kcal/mol, respectively. In Figure 7, the docked postures are shown. The results showed that when each receptor was docked with certified natural ligands, it had superior docked scores and binding energies than when the outcome was anticipated using a chemical equivalent. Pyrimethamine [5-(4-chlorophenyl)-6-ethylpyrimidine-2,4-diamine Chloridine], an FDA-approved chemical molecule, is highly selective against the proteins that cause dengue fever. Its efficacy against DENV has been previously documented [53]. As a result, it has been recommended that various natural ligands be used to attack certain infectious and dangerous targets. Furthermore, using natural substances to treat a variety of recently emerging infections has become a popular method in medicinal chemistry since these molecules are unlikely to induce adverse effects that would otherwise be induced by pharmaceuticals [54]. Moreover, these bioactive natural ligands are major components of widely available plants with significant therapeutic potential, which are still utilized in traditional medicine to treat a variety of viral infections [55].

### 2.4. Molecular Dynamic Simulation Analysis

The binding of a compound to the binding site of a protein can lead to observable conformational changes in the dynamics of the targeted protein. Root mean square deviation (RMSD) is one of the most important fundamental properties for establishing whether the protein is stable and close to the experimental structure [56] According to the RMSD plot, native, alepterolic acid, sphaeropsidin A, and stevioside binding kept the dynamics of targeted proteins at less than 0.3 nm, whereas triptolide binding resulted in more structural deviations from its native conformation (Figure 8A). In the case of the native-bound 1OKE and alepterolic acid, sphaeropsidin A, and stevioside-bound 1OKE complexes, the nature of their dynamics was the same during 100 ns of MD simulation.

In another case, the dynamics of caesalacetal, methyl dodovisate A, and stevioside-bound 2VBC were less than native-bound 2VBC, while the dynamics of sphaeropsidin A-bound 2VBC increased dramatically after 60 ns (Figure 8). For 4O6B, all of the selected compounds had a good dynamical effect on 4O6B, where all RMSD values of selected compound-bound 4O6B fluctuated less than 0.3 nm during the 100 nm. It can also be observed that caesalacetal and triptolide diminished the degree of fluctuation less than the native-bound 4O6B (Figure 8C).

For another targeted protein (4V0Q), the average value for the RMSD of native-bound 4V0Q was ≈ 0.287 as shown in Figure 8D. Further, we can observe that caesalacetal and stevioside reduced the dynamics of 4V0Q when they bound to it. Triptolide, on the other hand, increased the overall RMSD fluctuation by 100 ns more than the native ligand (Figure 8D). Moreover, the dynamics of understudied drugs inside the active site were compared and presented in Figure 8B. It can be observed that all of these ligands have nearly the same nature of movements inside the active site.

To investigate the dynamics of the protein’s backbone residues in the protein-ligand complexes compared to the Native-bound protein, the root means square fluctuations (RMSF) of the backbone atoms of the protein were depicted in Figure 9. Figure 9A reveals that the alepterolic acid, sphaeropsidin A, stevioside, and triptolide reduced the RMSF values of 1OKE when compared with native-bound protein. For the second target (2VBC) methyl dodovisate A and stevioside had a significant impact on increasing the fashion of fluctuation of RMSF of 2VBC when compared with native (Figure 9B). Whilst the third target (4O6B), the overall average RMSF value for native-bound 4O6B (Figure 9C), is higher than the caesalacetal, sphaeropsidin A, stevioside, and triptolide-bound 4O6B, and for the fourth target (4V0Q), as shown in Figure 9D, stevioside nearly fluctuated higher than the dynamics of native-bound 4V0Qand rest compound-bound 4V0Q complexes.

### 2.5. MM-PBSA Analysis

Hydrogen bond number and distribution in the selected targets with the selected compounds were studied to determine the stability of the protein-drug interactions inside the binding site during the 100 ns simulation period (Table 7). The hydrogen bond number results showed that stevioside had the highest number of hydrogen bonds (2.116988301) with 1OKE when compared with native, alepterolic acid, sphaeropsidin A, and triptolide Figure 10. Again, stevioside exhibited the highest average number of hydrogen bonds (3.206679332) compared to native, caesalacetal, methyl dodovisate A, and sphaeropsidin compounds as presented in Figure 10B. Further, stevioside, showed very strong interaction (average number of hydrogen bonds is 4.02439756) with 4OBE (Figure 10C). Most interestingly, the behavior in which stevioside again established the highest number of hydrogen bonds with 4V0Q (average is 2.765623438) is illustrated in Figure 10D. The hydrogen bond results help in understanding the functionality and ability of stevioside to work as a multi-target binder.

### 2.6. Network Pharmacology of Diterpenoids

#### 2.6.1. Gene Set Enrichment Analysis

Enrichment analysis is a versatile method to gain insight into the pathways whose activity is influenced by a particular gene group. As seen in Figure 11A, our biological process enrichment analysis indicated that the list of genes targeted by our medicinal plant compounds was most significantly associated with DNA metabolic process, DNA repair, and replication, cellular response to DNA damage stimulus, nucleocytoplasmic transport, regulation of translational initiation, protein localization to nuclear envelope, regulation of translation, and spliceosomal snRNP assembly regulation of cell cycle G_2_/M phase transition pathways. The top 10 molecular function enrichments are mismatched DNA binding, damaged DNA binding, ribosomal small subunit binding, nucleoside diphosphate kinase activity, phosphatidylinositol phospholipase C activity, phospholipase C activity, nucleobase-containing compound kinase activity, phosphoric diester hydrolase activity, double-stranded DNA binding, and snRNA binding, as shown in Figure 11B. Figure 11C illustrates the cellular components that include the spindle pole centrosome, methylosome, centrosome, small nuclear ribonucleoprotein complex, microtubule organizing center, U4/U6 x U5 tri-snRNP complex, replication fork, spliceosomal tri-snRNP complex, centriole, and nuclear body. The KEGG pathway annotation showed that pathways in cytosolic DNA-sensing pathway, longevity regulating pathway, prion diseases, epithelial cell signaling in *Helicobacter pylori* infection, tumor necrosis factor (TNF) signaling pathway, RNA transport, vasopressin-regulated water reabsorption, ErbB signaling pathway, spliceosome and cysteine, and methionine metabolism were at the top of the list, as shown in Figure 11D.

#### 2.6.2. Construction of “Drug-Target-Pathway” Network

The seven drug candidates, 313 gene targets, and the top 10 pathways were imported into Cytoscape (V.3.8.2) software, and the “Drug-Target-Pathway” network was obtained as shown in Figure 12. Light green hexagonal symbols represent the chosen drugs, the blue circle represents potential targets, and the red triangle represents pathways. Through the software, the node is visualized by degree value, and the node size is proportional to degree value. According to the requirements of topological parameters, the key nodes were determined by degree values greater than twice the median to obtain potential active components for later molecular docking tests.

## 3. Materials and Methods

### 3.1. Protein/Macromolecule Structure Preparation

The crystal structural protein of DENV envelope (E) protein (PDB ID: 1OKE, resolution: 2.40 Å) [57], a serine protease, and ATP-dependent helicase (NS3) (PDB ID: 2VBC. resolution: 3.15Å) [58], RNA-dependent RNA polymerase (NS5) (PDB ID: 4V0Q, resolution: 2.3 Å) [59], NS1 (PDB ID: 4O6B, resolution: 3 Å) [60] (Figure 1). Three-dimensional crystal enzyme structures in PDB format for Structural Bioinformatics were downloaded from the Protein Data Bank (PDB) (https://www.rcsb.org/, accessed on 1 June 2021) and for energy minimization in the crystal structure, we utilized the Swiss-PDB Viewer software package (version 4.1.0), and then all the heteroatoms and water molecules of proteins were removed by usingPyMOl (V.2.4.) before docking [61]. These structures were examined critically using Ramachandran Plot by ProCheck [62] to inspect the superior quality of the target protein structures selected for docking studies. All the crystallographic water molecules and associated heteroatoms were eliminated from the original crystal structures, and polar hydrogen atoms were added along with the Kollman charges. The geometry of the original moiety was rectified and visualized by PyMol (V.2.4) [63].

### 3.2. Active Site Prediction

For efficient docking, CASTp [24] has been used to approximate viral receptor active sites, and PyMol (V.2.4) was used to describe the Cartesian coordinates x, y, and z (active sites). Auto Dock Vina also used these regions to create grid boxes for molecular docking [64]. The active sites with the highest scores were characterized as a required precursor for the production of a grid in identified viral and vector receptors. CASTp was used to characterize and measure the active sites, binding sites, internal inaccessible cavities, surface accessible structural pockets and structure, and protein cavities [65].

### 3.3. Selection and Preparation of Ligands

There are 160 diterpenes/diterpenoids that were collected and selected from natural resources and mentioned by the literature-screening procedure [66,67,68]. From them, nearly 20 diterpenoids are available and showed anti-DENV activity in several in vivo experimental systems (Table 8) [23] as well as the FDA-approved drug, pyrimethamine, were obtained from the PubChem repository sample in the “sdf” file format. Pyrimethamine (Pubchem ID: 4993), a DENV NS2B/3 protease inhibitor, could block the translation and polyprotein processing in DENV [51], particularly at one intramolecular cleavage site within NS3 [52]. All internal energies of the ligands were optimized by using Chem3D Pro12.0 program packages [69]. The chemical structures were drawn using Chemsketch [70]. The final optimized and prepared ligands were used for molecular docking (Table 8).

### 3.4. In Silico Pharmacokinetic Study

For identification of drug likeliness of diterpene ligands, a study has been utilized similar to the way that Lipinski’s rule of five [71], Ghose filters [72], CMC 50 like rule [73], Veber filter [74], MDDR like rule [75], BBB likeness (Clark, 2003) [76], and QED [77] were used to gauge the compounds. Lipinski’s filter was used to measure hydrophobicity. Many of the ligands in the active compounds had their pharmacokinetic profile and physicochemical descriptors expected. In drug development, these guidelines have been used to preselect bioactive molecules [78].

ADME (Adsorption, Distribution, Metabolism, and Excretion) is important to analyze the pharmacodynamics of the proposed molecule that could be used as a drug. SWISS-ADME tool [79] is a website (https://www.swissadme.ch, accessed on 5 June 2021) which allows the user to draw their respective ligand or drug molecule or include SMILES data from PubChem and provides the parameters, such as lipophilicity (iLOGP, XLOGP3, WLOGP, MLOGP, SILICOS-IT, Log P0/w), water solubility-Log S (ESOL, Ali, SILICOS-IT), drug-likeness rules (Lipinski, Ghose, Veber, Egan, and Muegge) and Medicinal Chemistry (PAINS, Brenk, Leadlikeness, Synthetic accessibility) methods [79]. Data from PubChem, which consists of SMILES of diterpene ligands (https://pubchem.ncbi.nlm.nih.gov/compound, accessed on 5 June 2021) was entered into the search bar and was analyzed.

Toxicology prediction of small molecules is important to predict the tolerability of the small molecules before being ingested by human and animal models. pkCSM is an online database in which the small molecule can be drawn virtually or can be analyzed by submitting the SMILES of the same. The website can provide details of toxicology effects in the fields of human maximum tolerated dose, LD_50_, hepatotoxicity, and Minnow toxicity. The website was logged on and SMILES of the diterpene ligands data from PubChem was searched and submitted into the website, and toxicity mode was selected [80].

### 3.5. Molecular Docking Protocol

In medicinal chemistry, molecular docking is a numerical tool for drug design. The Auto Dock Vina tool uses this approach to predict the pharmacodynamic profile of drug candidates by ranking and orienting them to receptor binding sites [81]. The docking outcome specifies the degree of ligand interaction with the desired protein’s active site. The active binding sites of the target protein are the locations of the ligand in the initial target protein grids (40 × 40 × 40) [82], with PyMol, Auto dock Vina, and Drug Discovery Studio (v.20.1.0.19295) being used to examine them [83].

### 3.6. Molecular Dynamics (MD) Simulation Study

MD modeling is now considered a decisive step in computer-aided research for drug discovery at the atomic level. By studying the internal movement of proteins, many mysterious biological functions of proteins and their deep dynamic mechanisms can be revealed [56]. Regarding the dynamically changing time scale, we can use this time scale to judge whether the protein-ligand complex is stable [84]. In this study, we performed MD simulations on the four proteins with docked ligands in addition to cocrystal ligands, produced by molecular binding on a time scale of 100 ns. We used the GROMACS 2018.1 package [85] to run the MD simulation. The CHARMM 27 force field [86] was used to parameterize the ligand-protein complex of all atoms. The intermolecular three-point transfer potential (TIP3P) was chosen as the solvent [87], adding Na + or Cl-ions to adjust the charge to simulate the physiological environment. Then we used the steepest descent algorithm [56] with an allowable value of 1000 kJ/mol·nm to minimize the energy of these systems. In the next step, the NVT and NPT pools canceled out the positionally restricted complexes on the protein molecule within 0.1 ns. Then, MD simulations with no restrictions on protein molecules or ligands were performed to determine the stability within 100 ns. Finally, some Gromacs modules were used to analyze MD trajectories, such as gmx rms, gmxrmsf, and gmxhbond.

### 3.7. Molecular Mechanics Poisson–Boltzmann Surface Area (MMPBSA) Analysis

We utilized the g_mmpbsa tool to estimate the binding free energies of the protein-ligand systems [88]. One of the popular methods to estimate the interaction energies are Molecular Mechanics Poisson–Boltzmann Surface Area (MMPBSA) analysis. This method uses molecular dynamics simulation trajectories to predict binding free energies (ΔEMMPBSA) of protein-protein, protein-ligand, or protein–DNA systems. We performed the MMPBSA analysis on the last 20 ns of the MD trajectory of each protein-ligand system at an interval of 50 ps. Total binding energies of the protein-ligand complexes are presented in Table 1. It can be observed that binding free energies of the chosen compounds with selected targeted proteins alter between different values, owing to differences in the mode of binding. However, the most promising results which can be obtained from this table are those related to binding affinities of stevioside with viral envelope (E) protein (PDB ID: 1OKE), serine protease (NS3) protein (PDB ID: 2VBC), RNA-directed RNA polymerase (NS5) (PDB ID: 4V0Q), and non-structural protein 1 (NS1) (PDB ID: 4O6B).

### 3.8. Network Pharmacology of Diterpenoid

Three main stages are included in network pharmacology assessment: (a) target estimation for selected bioactive ligands; (b) amplification study for predicted targets, and (c) network construction and analysis for selected ligands, targets, and pathways. Briefly, targets of chosen drugs were predicted using DIGEP-Pred [89] at the pharmacological activity (Pa) of 0.5. Next, the predicted proteins of chosen drugs were enriched using STRING (V.11.0) [90] to generate the protein-protein interaction. All proteins which are obtained from the STRING database were submitted to the Enricher database (https://maayanlab.cloud/Enrichr/, accessed on 9 July 2021) to enrich their biological processes, molecular function cellular components, and KEGG pathways. Finally, the network between chosen drugs, their targets, and pathways was constructed using Cytoscape(V.3.8.2.) [91].

## 4. Conclusions and Final Considerations

Dengue fever is a severe infectious endemic illness that affects over 100 countries across the globe, from Africa to the Western Pacific. It is caused by DENV, which is transmitted to people by an *A. aegypti* mosquito bite. Dengue fever and dengue hemorrhagic fever have killed millions of people across the world. Natural phytocompounds have a potential anti-viral effect. Among them, diterpenes/diterpenoids are the most prominent bioactive lead compounds. In this study, among the selected drug candidates, triptolide, stevioside, alepterolic acid, sphaeropsidin, methyl dodovisate, andrographolide, caesalacetal, and pyrimethamine have good to moderate binding affinities compared with the FDA-approved anti-viral medication (pyrimethamine). Our findings will benefit future nonclinical, preclinical, and clinical investigations using these compounds, as well as encourage medicinal chemistry specialists to perform relevant studies on these potential natural lead compounds and their derivatives.

## Figures and Tables

**Figure 1 molecules-26-06821-f001:**
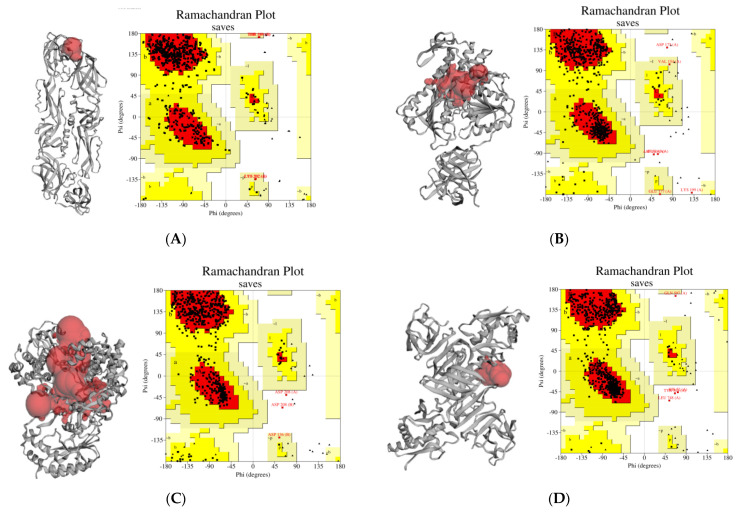
The estimated active sites, which make up the amino acids, are shown in the active site identification (red pocket) findings from the CASTp network and structure validation (by Procheck). (**A**) Viral envelope (E) protein (PDB ID: 1OKE); (**B**) serine protease (NS3) protein (PDB ID: 2VBC); (**C**) RNA-directed RNA polymerase (NS5) (PDB ID: 4V0Q); (**D**) non-structural protein 1(NS1) (PDB ID: 4O6B). [Some errors (letters in Ramachandran plot) are generated by automated software which can’t be changed maually].

**Figure 2 molecules-26-06821-f002:**
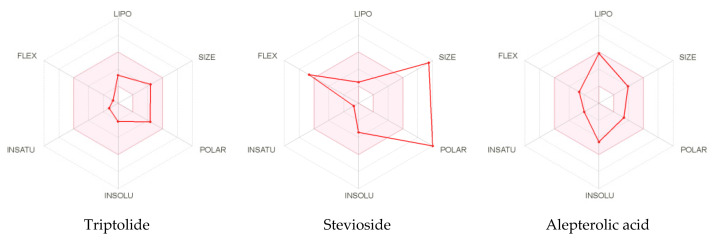
Summary of physiochemical, pharmacokinetics, and toxicological properties of selected ligand candidates (the color space is a suitable physiochemical space for oral bioavailability; **LIPO** (**Lipophility**): –0.7 < XLOGP3 < 5.0; **SIZE**: 150 g/mol < MW < 500 g/mol; **POLAR** (**Polarity**): 20 Å^2^ < TPSA < 130 Å^2^; **INSOLU** (**insolubility**): 0 < LogS(ESOL) < 6; **INSATU** (**in saturation**): 0.25 < FractionCsp3 < 1; **FLEX** (**Flexibity**): 0 < Num. rotatable bonds < 9).

**Figure 3 molecules-26-06821-f003:**
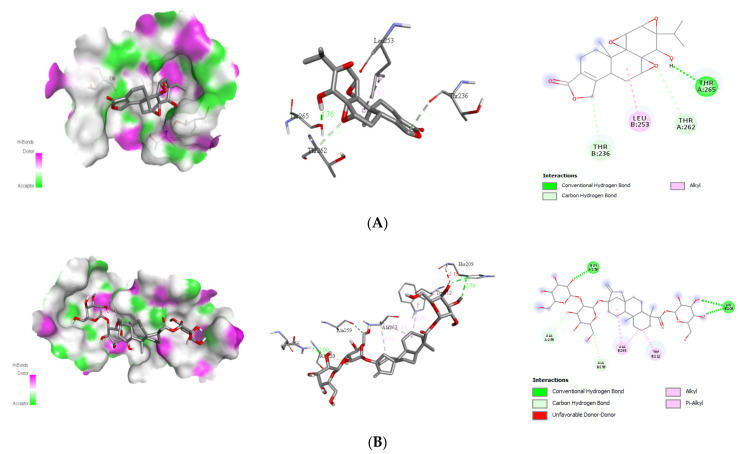
Binding poses of four top-ranked compounds at the binding site of the dengue virus envelope (E) protein (PDB ID: 1OKE) and 2D and 3D interaction diagrams. (**A**) triptolide-E protein; (**B**) stevioside-E protein; (**C**) alepterolic acid-E protein, and (**D**) sphaeropsidin A-E protein.

**Figure 4 molecules-26-06821-f004:**
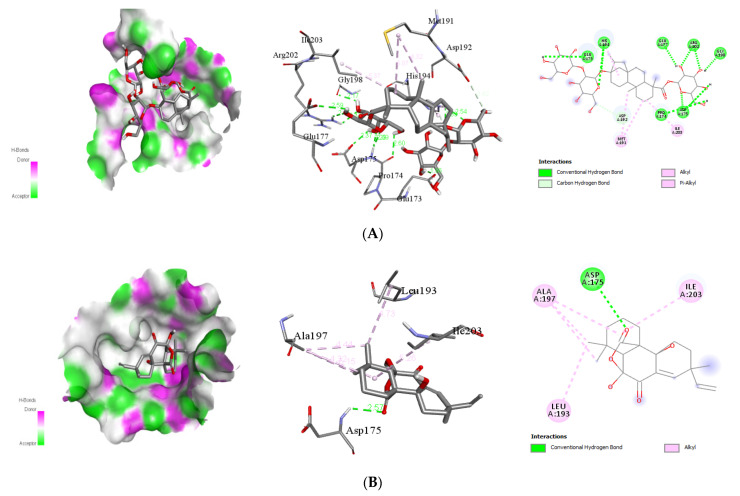
Binding poses of four top-ranked compounds at the binding site of dengue viral NS3 (PDB ID: 2VBC) and 2D and 3D interaction diagrams. (**A**) Stevioside-NS3; (**B**)sphaeropsidin A-NS3; (**C**) methyl dodovisate A-NS3, and (**D**) caesalacetal-NS3.

**Figure 5 molecules-26-06821-f005:**
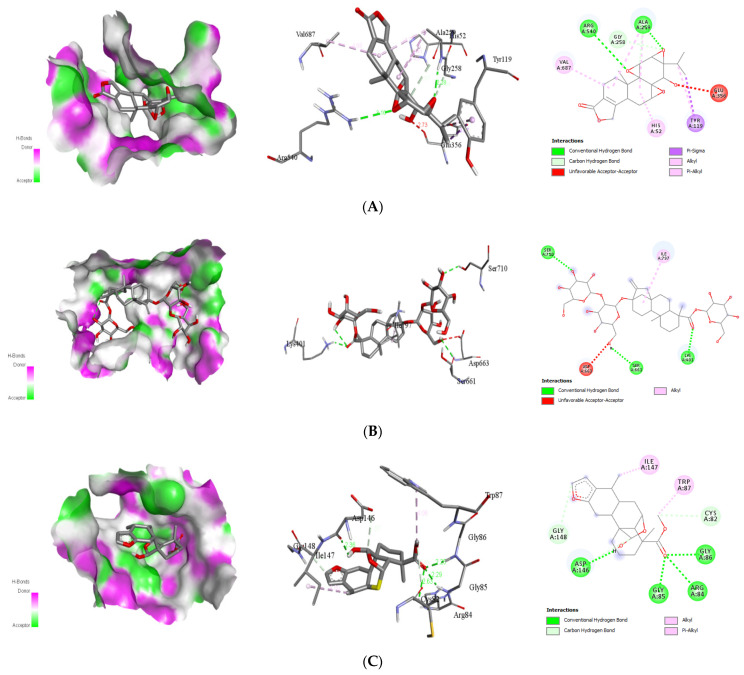
Binding poses of four top-ranked compounds at the binding site of dengue virus NS5 (PDB ID: 4V0Q) and 2D and 3D interaction diagrams. (**A**) Triptolide-NS5; (**B**) stevioside-NS5; (**C**) caesalacetal-NS5; (**D**) andrographolide-NS5.

**Figure 6 molecules-26-06821-f006:**
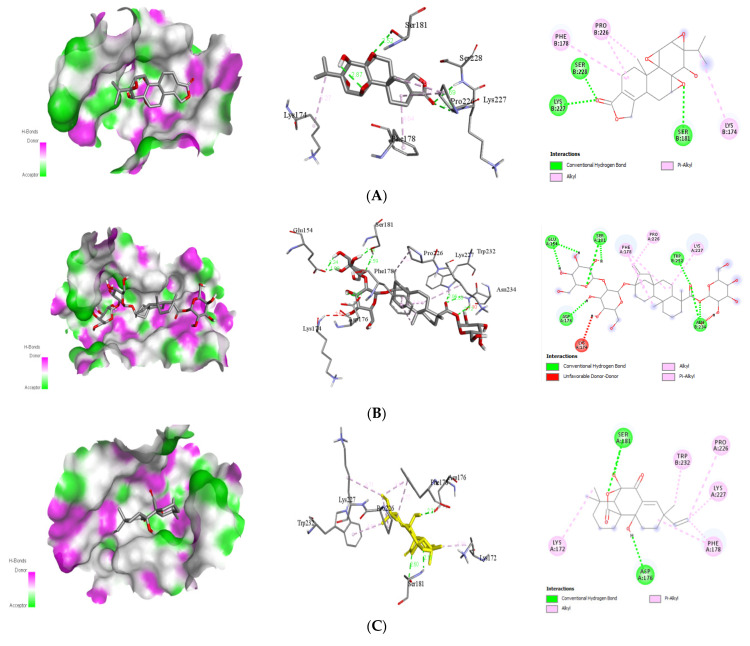
Binding poses of four top-ranked compounds at the binding site of dengue virus NS1 (PDB ID: 4O6B) and 2D and 3D interaction diagrams. (**A**) Triptolide-NS1; (**B**) stevioside-NS1; (**C**)sphaeropsidin A-NS1; (**D**) caesalacetal-NS1.

**Figure 7 molecules-26-06821-f007:**
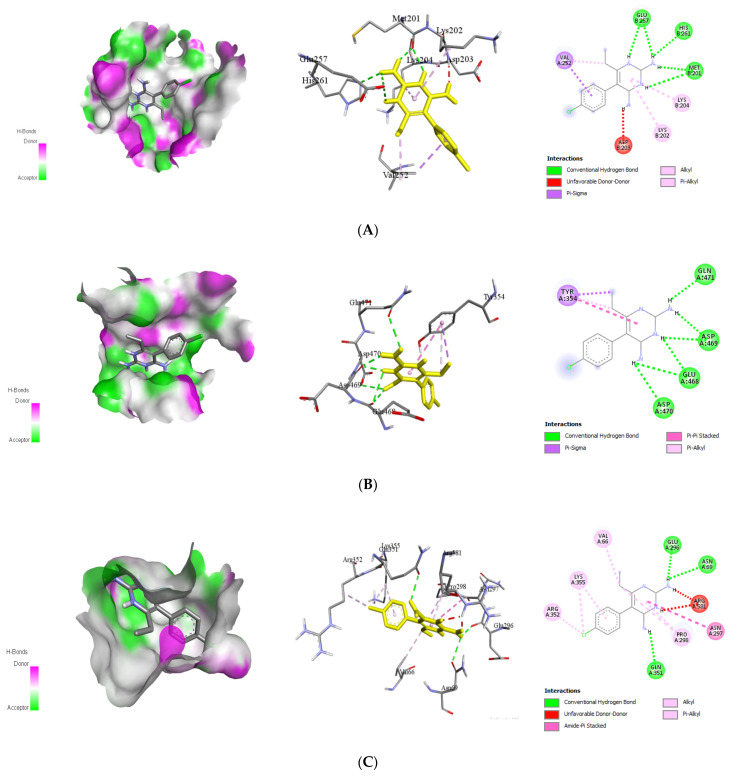
Interaction of reference drugs (pyrimethamine; IUPAC name: 5-(4-chlorophenyl)-6-ethylpyrimidine-2,4-diamine-chloridine) with dengue virus protein. (**A**) Envelope (E) protein (PDB ID: 1OKE); (**B**) NS3 (PDB ID: 2VBC); (**C**) NS5 (PDB ID: 4V0Q); (**D**) NS1 (PDB ID: 4O6B).

**Figure 8 molecules-26-06821-f008:**
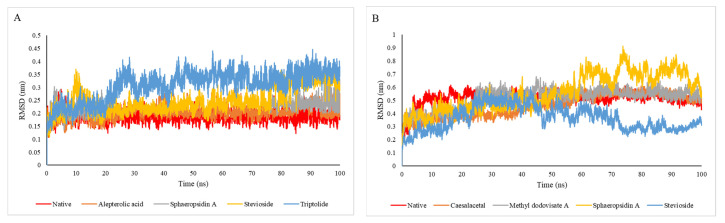
The RMSD plot for the backbone atoms for (**A**) 1OKE; (**B**) 2VBC; (**C**) 4O6B, and (**D**) 4V0Q.

**Figure 9 molecules-26-06821-f009:**
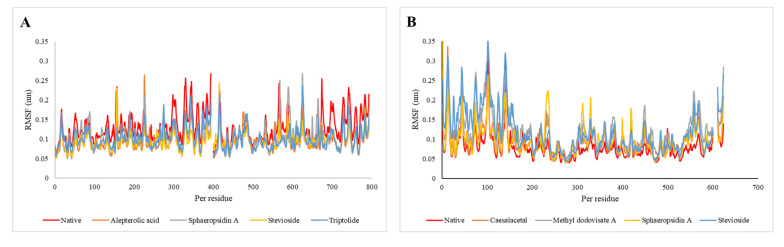
The RMSF plot for the backbone atoms for (**A**) 1OKE, (**B**) 2VBC, (**C**) 4O6 Band (**D**) 4V0Q.

**Figure 10 molecules-26-06821-f010:**
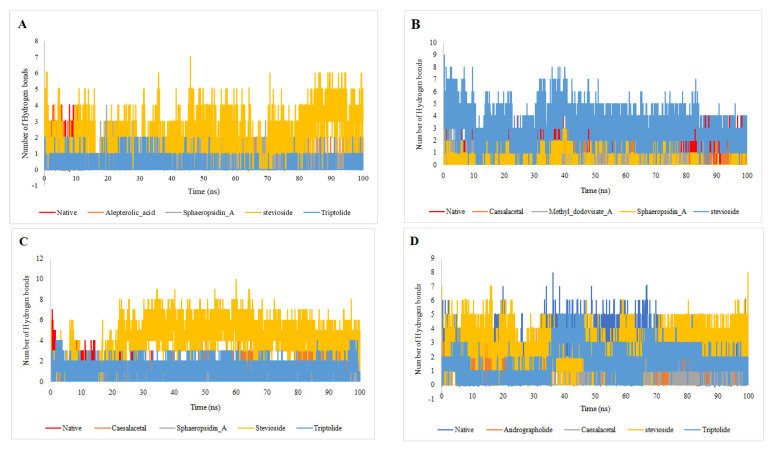
The hydrogen bonds plot between selected compounds with (**A**) 1OKE; (**B**) 2VBC; (**C**) 4O6B, and (**D**) 4V0Q.

**Figure 11 molecules-26-06821-f011:**
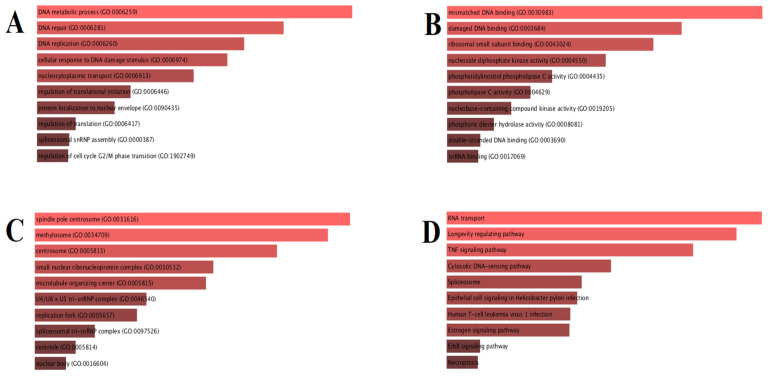
Gene ontology (GO) functional and Kyoto Encyclopedia of Genes and Genomes (KEGG) pathway enrichment analysis for the selected genes (performed via enricher) (**A**) GO Biological Process 2018; (**B**) GO Molecular Function 2018; (**C**) GO Cellular Component 2018; (**D**) KEGG 2019.

**Figure 12 molecules-26-06821-f012:**
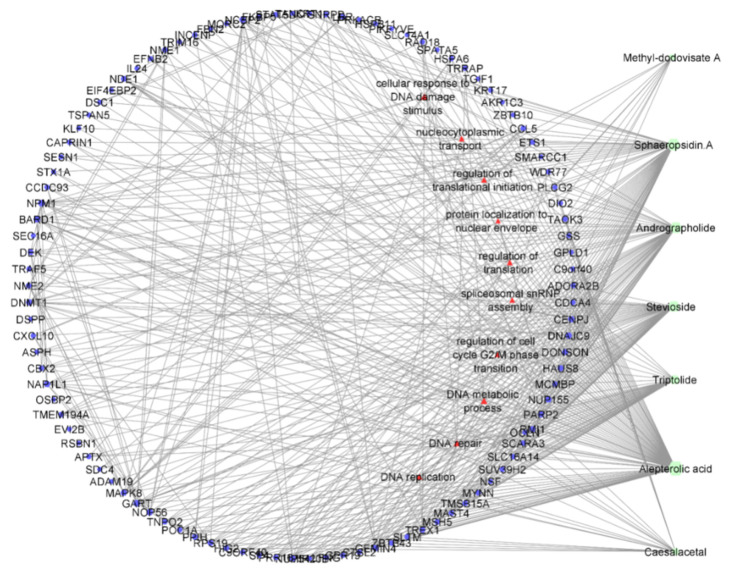
Drug-targets-pathway networks.

**Table 1 molecules-26-06821-t001:** ADMET and drug likeliness properties for molecules.

Properties	Triptolide	Stevioside	AlepterolicAcid	Sphaeropsidin A	Methyl DodovisateA	Andrographolide	Caesalacetal	Pyrimethamine
**Formula**	C20H24O6	C38H60O18	C20H32O3	C20H26O5	C22H28O2	C20H30O5	C21H28O5	C12H13ClN4
**Molecular weight (g/mol)**	360.40	804.87	320.47	346.42	324.46	350.45	360.44	248.71g/mol
**H-Bond Acceptors**	6	18	3	5	2	5	5	2
**H-Bond Donors**	1	11	2	2	0	3	1	2
**Num. Rotatable Bonds**	1	10	4	1	5	3	2	2
**TPSA (Å^2^)**	84.12	294.98	57.53	83.83	26.30	86.99	68.90	77.82
**Fraction Csp3**	0.85	0.92	0.75	0.70	0.50	0.75	0.76	0.17
**Molar Refractivity**	88.54	188.26	95.49	91.95	100.18	95.21	95.30	71.06
**LogPo/w (XLOGP3)**	0.22	−1.20	4.74	2.64	5.31	2.16	3.08	2.69
**LogS (ESOL)**	−2.15	−3.41	−4.55	−3.58	−4.87	−3.18	−4.03	−3.47
**Max. tolerated dose (human) (logmg/kg/day)**	−0.321	−1.524	−0.297	−0.074	−0.159	0.128	−0.14	0.113
**Oral Rat Acute Toxicity (LD50; mol/kg)**	3.107	2.597	2.28	1.92	1.779	2.162	2.581	2.912
**Hepatotoxicity**	No	No	No	No	No	No	Yes	No
**Minnow toxicity (logmM)**	1.983	9.202	0.459	1.606	−0.525	1.37	0.418	0.919
**Blood brain barrier (logBB)**	−0.362	−2.029	−0.018	0.016	0.629	−0.598	−0.163	0.278
**HIA (%)**	83.195	0	94.672	95.753	97.808	95.357	97.564	92.738
**CaCo2 Permeability**	0.401	−1.087	1.432	1.135	1.64	1.07	1.145	0.927
**Total Clearance** **(logml/min/kg)**	0.484	0.691	1.122	0.541	1.381	1.183	0.538	−0.033

**Table 2 molecules-26-06821-t002:** The four best results for the docking of natural bioactive ligands with viral envelope (E) protein (PDB ID: 1OKE) proteins target.

Compounds	Target	Interact Residues	No. of H-Bond	H-Bond Residues	H-Bond Length	Binding Energy (kcal/mol)
Triptolide	1OKE	Leu253 Thr236 Thr262	1	Thr265	1.76	−8.1
Stevioside		Ala259 Ala263 Trp212	2	Gln256 Hios209	2.09 2.16	−8.4
Alepterolic acid		Leu253 Pro217	2	Gln256 Thr265	2.31 1.87	−8.3
Sphaeropsidin A		His261 Thr265 Trp206	0	-	-	−8.7

**Table 3 molecules-26-06821-t003:** The four best results for the docking of natural bioactive ligands with viral proteins target.

Compounds	Target	Interact Residues	No. of H-Bond	H-Bond Residue	H-Bond Length	Binding Energy (kcal/mol)
Stevioside		Asp192 Ile203 Met191 His194	7	Arg202 Asn175 Glu173 Glu177 Gly198 His194 Pro174	2.33 2.29 1.96 2.59 2.17 2.39 2.60	−8.0
Sphaeropsidin A	2VBC	Ala197 Ile203 Leu193	1	Asp175	2.57	−8.3
Methyldodovisate A		Asp258 Arg215 Arg217 His251 Ile256	3	Arg254 Gly253 Thr252	2.17 2.17 2.94	−9.2
Caesalacetal		Ala197 His194 Leu193	1	Asp175	2.59	−8.0

**Table 4 molecules-26-06821-t004:** The four best results for the docking of natural bioactive ligands with viral NS5 proteins target.

Compounds	Target	Interact Residues	No. of H-Bond	H-Bond Residues	H-Bond Length	Binding Energy (kcal/mol)
Triptolide		Glu356 Gly258 His52 Tyr119 Val687	2	Ala259 Arg540	2.28 3.10	−8.8
Stevioside	4V0Q	Asp663 Ile797	3	Lys401 Ser661 Ser710	2.23 2.83 2.53	−9.4
Andrographolide		Ile797 Val603	4	Asp663 Gly604 Thr605 Tyr606	2.25 2.39 2.56 1.82	−8.4
Caesalacetal		Cys82 Gly148 Ile147 Trp87	4	Arg84 Asp146 Gly85 Gly86	2.63 2.36 2.29 2.25	−8.4

**Table 5 molecules-26-06821-t005:** The four best results for the docking of natural bioactive ligands with viral proteins target (NS1).

Compounds	Target	Interacting Residues	No. of H-Bond	H-Bond Residue	Bond Length (A)	Binding Energy (Kcal/mol)
Triptolide		Lys174 Phe178 Pro226	3	Lys227 Ser181 Ser227	2.28 2.52 2.69	−8.3
Stevioside		Lys174 Lys227 Phe178 Pro226	5	Asn234 Asp176 Glu154 Ser181 Trp232	2.72 2.41 1.99 2.14 2.06	−9.3
Sphaeropsidin A	4O6B	Lys172 Lys227 Phe178 Pro226 Trp232	2	Asp176 Ser181	2.21 2.16	−8.5
Caesalacetal		Glu173 Lys227 Phe178 Ser181 Trp232	2	Ser228 Trp210	2.33 2.39	−8.5

**Table 6 molecules-26-06821-t006:** Results for the docking of pyrimethamine with all four dengue viral protein target proteins.

Compounds	Target	Interacting Residues	No. of H-Bond	H-Bond Residue	Bond Length (Å)	Binding Energy (kcal/mol)
Pyrimethamine	E protein (1OKE)	Asp203 Lys202 Lys204 Val252	5	Glu257 His261 Met201	2.55 2.60 2.44	−7.5
	NS3(2VBC)	Tyr354	6	Asp469 Asp470 Gln471 Glu468	2.15 2.83 2.56 2.63	−6.3
	NS5(4V0Q)	Arg352, Arg581, Asn297, Lys355, Pro298, Val66	3	Glu296 Asn69 Glu351	2.04 2.60 2.55	−7.8
	NS1(4O6B)	Phe178 Ser181	3	Asp176 Asp180 Cys179	2.32 2.42 2.43	−6.6

**Table 7 molecules-26-06821-t007:** Calculation of MMPBSA energy (ΔEMMPBSA) of the selected proteins and the nominated ligands.

Target Name	Compound Name	Vander Waals Energy (kJ/mol)	Electrostatic Energy (kJ/mol)	Polar Solvation Energy (kJ/mol)	SASA Energy (kJ/mol)	Binding Energy (kJ/mol)
1OKE	Native	−96.764	314.382	−253.723	32.344	−3.761
	Alepterolicacid	−88.371	−14.159	68.7155	−12.475	−46.2895
	SphaeropsidinA	−122.068	−5.756	71.871	−14.922	−70.875
	Stevioside	−83.430	−12.410	73.405	−11.170	−33.605
	Triptolide	−130.551	−13.742	66.847	−13.472	−90.918
2VBC	Native	−149.888	−15.821	87.180	−16.847	−94.556
	Caesalacetal	−148.616	−24.435	104.966	−16.718	−84.309
	MethyldodovisateA	−138.175	−14.924	118.165	−14.918	−49.852
	SphaeropsidinA	−115.183	3.982	43.248	−13.188	−81.141
	Stevioside	−195.236	−22.923	144.028	−22.388	−96.519
4O6B	Native	−60.911	−32.869	89.094	−9.556	−14.602
	Caesalacetal	−139.283	−33.117	81.796	−13.614	−104.281
	SphaeropsidinA	−90.141	−8.125	43.680	−9.868	−64.454
	Stevioside	−162.844	−110.614	151.263	−22.976	−145.171
	Triptolide	−116.724	−75.279	100.616	−13.566	−104.953
4V0Q	Native	−135.681	−41.363	131.823	−14.601	−59.822
	Andrographolide	−0.036	4.881	58.619	−2.893	60.571
	Caesalacetal	−108.493	3.610	35.992	−12.449	−81.34
	Stevioside	−270.746	−68.928	236.892	−28.205	−130.987
	Triptolide	−178.301	−24.621	210.824	−19.300	−11.398

**Table 8 molecules-26-06821-t008:** Diterpenes/diterpenoids and their derivatives with bioactivity against DENV and DENV-infected experimental animals or cell lines.

Compounds	Source	PubChem ID	Chemical Structure
Phorbol ester	*Jatropha curcas*	22833501	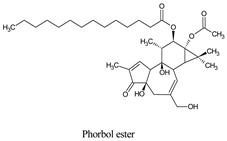
Triptolide	*Tripterygium wilfordii*	107985	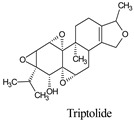
Steviol	*Stevia rebaudiana*	452967	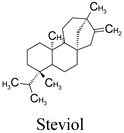
Ferruginol	*Prumnopitys andina*	442027	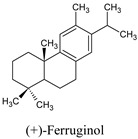
18-oxoferruginol	*Torreya nucifera*	52946772	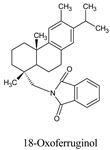
Andrographolide	*Andrographis paniculata*	5318517	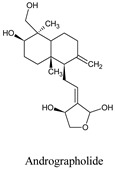
Stevioside	*Stevia rebaudiana*	442089	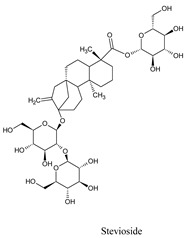
Rebaudioside A	*Stevia rebaudiana Bertoni*	6918840	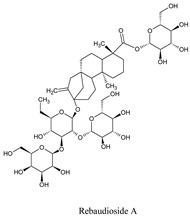
Forskolin	*Coleus forskohli*	47936	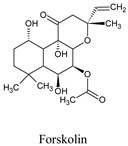
Ent-kaur-16-en-19-oic acid	*Elaeoselinum foetidum*	73062	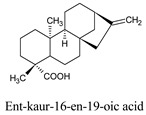
9(11),16-kauradien-19-oic acid	*Melantheria albinervia*	14635430	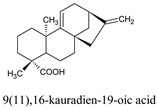
Alepterolic acid	*Copaifera reticulata*	13858188	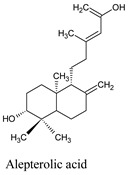
Zanzibaric acid	*Hymenaea courbaril*	101289556	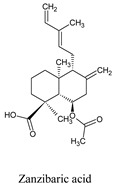
Isoozic acid	*Hymenaea courbaril*	100983062	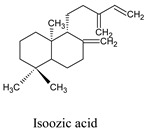
Sphaeropsidin A	*Diplodia cupressi*	51361447	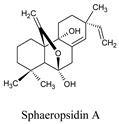
Phytol	*Hierochloëo dorata*	5280435	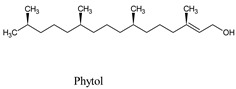
7-oxo-8,11,13-cleistanthatrien-3-ol	*V. gigantea*	132609177	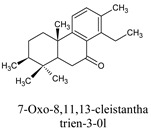
Methyl dodovisate A	*Hierochlo ëodorata*	146156767	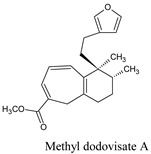
Caesalacetal	*Caesalpinia decapetala* var.	132918611	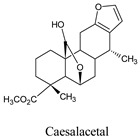
Caesaljapin	*Caesalpinia decapetala* var.	6712179	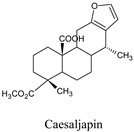

## Data Availability

Not applicable.

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
