# Peer review of "Diterpenes/Diterpenoids and Their Derivatives as Potential Bioactive Leads against Dengue Virus: A Computational and Network Pharmacology Study"

_molecules, 2021, doi:10.3390/molecules26226821_

Round 1

Reviewer 1 Report

Comments,

The manuscript by Hossian et al investigated the bioactivity against the dengue virus of diterpenes/ diterpenoids and their derivatives by a computational docking and pharmacology study. The results presented in the manuscript are very interesting and attract an audience in the field. Overall the manuscript is written well and scientifically strong. Thus, the manuscript is acceptable for publication in molecule journals, after considering the following points.

There are several typographic errors in the manuscript and should be properly addressed.

  1.  For example, some references are not properly written, and there are several types.
  2. Please provide the proper numbering for all diterpenoids derivatives in the tables.

Author Response

As mentioned in the file, the authors have corrected all the points which are addressed by the Reviewer. 

Reviewer 2 Report

In my opinion the manuscript is well written however discussion need to improve with recent work done in the specific to this manuscript. Introduction need to be little short.

In my previous evaluation am still in opinion to cut short the introduction portion and also clear demarcation between different parts

Abstract, Introduction, Materials and Methods , Results, Discussion and Conclusion. (Currrently intermingling)

Secondly Authors have used many Diterpenoids and their derivatives. These include triptolide, stevioside, alepterolic acid, sphaeropsidin A, methyl dodovisate A, andrographolide, caesalacetal, and pyrimethamine.

But authors have given too general statement that these all have anti dengue virus effects more as compared with FDA approved substance. But my suggestion is to quantify the effects of these all substances and make a comprehensive comparison of the efficacy of all these above mentioned Diterpenoids and their derivatives (numerically). The authors should finally suggest on these basis which one is the most efficacious I think in this way it will be more suitable and make this manuscript more interesting. 

Author Response

The authors have corrected the all points which are mentioned by the Reviewer. 
